# Metastatic Renal Medullary and Collecting Duct Carcinoma in the Era of Antiangiogenic and Immune Checkpoint Inhibitors: A Multicentric Retrospective Study

**DOI:** 10.3390/cancers14071678

**Published:** 2022-03-25

**Authors:** Zoé Guillaume, Emeline Colomba, Jonathan Thouvenin, Carolina Saldana, Luca Campedel, Clément Dumont, Brigitte Laguerre, Denis Maillet, Cécile Vicier, Frédéric Rolland, Delphine Borchiellini, Philippe Barthelemy, Laurence Albiges, Edouard Auclin, Matthieu Roulleaux Dugage, Stéphane Oudard, Constance Thibault

**Affiliations:** 1Medical Oncology, European Georges Pompidou Hospital, Université Paris Cité, 75006 Paris, France; zoe.guillaume@aphp.fr (Z.G.); edouard.auclin@aphp.fr (E.A.); matthieu.roulleaux-dugage@gustaveroussy.fr (M.R.D.); stephane.oudard@aphp.fr (S.O.); 2Medical Oncology, Gustave Roussy Institute, Université Paris-Saclay, 94805 Villejuif, France; emeline.colomba-blameble@gustaveroussy.fr (E.C.); laurence.albiges@gustaveroussy.fr (L.A.); 3Medical Oncology, Institut de Cancérologie Strasbourg Europe, 67200 Strasbourg, France; jonathan.thouvenin@chu-lyon.fr (J.T.); p.barthelemy@icans.eu (P.B.); 4Medical Oncology, Hopital Henri-Mondor, 94010 Créteil, France; carolina.saldana@aphp.fr; 5Medical Oncology, Pitié Salpêtrière Hospital, Sorbonne Université, 75013 Paris, France; luca.campedel@aphp.fr; 6Medical Oncology, Hôpital Saint Louis, Oncology Unit, 1 Avenue Claude Vellefaux, 75010 Paris, France; clement.dumont@aphp.fr; 7Medical Oncology, Centre Eugene—Marquis, 35000 Rennes, France; b.laguerre@rennes.unicancer.fr; 8Medical Oncology, Centre Hospitalier Lyon Sud, 69310 Pierre Bénite, France; denis.maillet@chu-lyon.fr; 9Medical Oncology, Institut de Cancérologie de l’Ouest, 44805 Saint-Herblain, France; vicierc@ipc.unicancer.fr (C.V.); frederic.rolland@ico.unicancer.fr (F.R.); 10Medical Oncology Department, Centre Antoine Lacassagne, Université Côte d’Azur, 06100 Nice, France; delphine.borchiellini@nice.unicancer.fr

**Keywords:** collecting duct carcinoma, metastatic renal medullary, Bellini carcinoma, immune checkpoint inhibitors, tyrosine kinase inhibitors

## Abstract

**Simple Summary:**

Collecting duct carcinoma (CDC) and renal medullary carcinoma (RMC) are two rare cancers with a very poor prognosis. Currently, first-line treatment is a platinum-based doublet, but very few studies have evaluated the effectiveness of treatments for subsequent lines. Additionally, despite the advent of new therapies in renal clear cell carcinoma, data are lacking on these types of cancers. Thus, we conducted a retrospective study in 11 centers in France to evaluate the different types and effectiveness of treatments received beyond first-line treatment in patients with metastatic CDC or RMC. Subsequent treatments showed limited efficacy regardless of the type of therapy received. This study supports the importance of finding therapeutic targets and/or biomarkers to improve patient outcomes.

**Abstract:**

Collecting duct carcinoma (CDC) and renal medullary carcinoma (RMC) are two rare subtypes of kidney cancer with a poor prognosis in the metastatic setting. Beyond first-line treatment, there are no standard-of-care therapies. This retrospective study assessed the efficacy of treatments after first-line chemotherapy in 57 patients with metastatic (m) CDC (*n* = 35) or RMC (*n* = 22) treated between 2010 and 2019 at 11 French centers. The median age was 53 years; overall, 60% (*n* = 34) of patients were metastatic at diagnosis. After a median follow-up of 13 months, the median overall survival was 12 (95% CI, 11–16) months. All patients received first-line platinum chemotherapy ± bevacizumab, with a median time to progression of 7.27 (95% CI, 7–100 months and an objective response rate (ORR) of 39% (95% CI, 26–52%). Patients received a median of two (1–5) treatment lines. Subsequent treatments included tyrosine kinase inhibitors (*n* = 12), chemotherapy (*n* = 34), and checkpoint inhibitors (*n* = 20), with ORR ranging 10–15% and disease control rates ranging 24–50%. The duration of response for all treatments was ~2 months. Notably, nine patients with CDC were still alive > two years after metastatic diagnosis. Beyond first-line therapy, treatments showed very low antitumor activity in mCDC/RMC. A better understanding of the biology of those rare tumors is urgently needed in order to identify potential targets.

## 1. Introduction

Collecting duct carcinoma (CDC, also known as Bellini tumor) and renal medullary carcinoma (RMC) are two extremely rare subtypes of renal cell carcinoma (RCC), accounting for less than 1% of renal cancers [1]. CDC is mostly diagnosed in men during the fifth decade of life, whereas RMC typically occurs in men during the second and third decades of life and is almost exclusively associated with a sickle-cell trait. Due to their aggressiveness, CDC and RMC are usually diagnosed at an advanced stage with symptomatic disease. Both entities are of poor prognosis, with a median overall survival (OS) of less than one year in the metastatic setting [2]. Both of them arise from the collecting duct epithelium of the kidney and share histological and macroscopic appearance, but RMC usually presents a loss of INI1 expression, due to a copy number alteration of *SMARCB1*.

For patients with localized CDC/RMC, the standard treatment remains nephrectomy [3]. In the metastatic setting, conventional systemic therapies used in the treatment of clear cell RCC (such as antiangiogenic, interferon-alpha/gamma, interleukin 2) have shown limited antitumor activity against CDC/RMC [3]. Recommended first-line treatment is a platinum plus gemcitabine chemotherapy regimen (CG regimen) based on the results of a prospective phase II trial conducted by the French Genito-Urinary Group (GETUG). The median progression-free survival (PFS) was 7.1 months, and the median OS was 10.5 months [4]. Recently, the prospective phase 2 BONSAI trial assessed cabozantinib, a multi-targeted tyrosine kinase inhibitor (TKI) of vascular endothelial growth factor receptor (VEGFR), MET, and AXL, showing promising results in the first-line treatment of metastatic CDC, with an objective response rate of 35% and a median PFS of 6.0 months [5].

However, few studies have evaluated subsequent treatments, and no standard of care has been established. Available data supporting the use of TKI or immune checkpoint inhibitors (ICIs) as second- or later-line treatments of metastatic CDC/RMC are mainly derived from case reports. Unfortunately, larger studies evaluating subsequent treatments are lacking. Therefore, we aimed to assess the efficacy of systemic treatments in patients with metastatic CDC/RMC previously treated with platinum-based chemotherapy.

## 2. Materials and Methods

We performed a retrospective multicenter study in 11 French genitourinary oncology expert centers between January 2010 and December 2019. To be eligible for inclusion, patients aged ≥ 18 years had to have histologically proven CDC/RMC and to have received at least one line of systemic treatment in the metastatic setting. Notably, patients enrolled in clinical trials of first- or subsequent-line therapies, including those treated with gemcitabine and cisplatin plus bevacizumab in the GETUG/FU24 trial (BEVABEL), were included. Data were collected from medical records by local investigators at each institution, through a uniform database template including baseline characteristics, clinical outcome, and safety at each line.

The primary outcome of the study was ORR of subsequent therapy lines, defined as the proportion of patients with a complete (CR) or partial response (PR) as the best response in accordance with Response Evaluation Criteria in Solid Tumors (RECIST) version 1.1. Secondary outcomes were OS, calculated from metastatic diagnosis until death or last follow-up, in the overall population and in the specific CDC and RMC populations, time to progression (TTP, defined as the time from treatment initiation until disease progression or death), disease control rate (DCR, defined as the proportion of patients with CR, PR, or stable disease (SD)), and the duration of treatment.

All variables were expressed as median (interquartile range) for quantitative variables or as numbers and percentages for qualitative variables. Overall survival and TTP were estimated using the Kaplan–Meier method. Statistical analyses and survival curves were performed using R version 4 software.

The study was approved by the Institutional Ethics Committee of APHP (#00011928, 2019-12-11), and patients still alive were informed.

## 3. Results

### 3.1. Population

In total, 57 patients were included across the 11 participating centers, including 35 (61%) patients with CDC and 22 (39%) patients with RMC. Median ages were 61 years and 33 years for CDC and RMC, respectively. The majority of patients were men (70%, *n* = 40) for both subtypes. All patients had metastatic disease, with most (60%, *n* = 34) presenting with metastases at diagnosis. Patient demographics and disease characteristics are provided in Table 1. The median number of treatment lines received was two (1–5).

At the time of analysis, 48 (84%) patients had died, and 2 patients were lost to follow up. After a median follow-up duration of 13 (95% CI, 10–15) months, the median OS was 12 (95% CI, 11–16) months. There was no difference between the two histological subtypes of 13 (95% CI, 10–18) months and 11 (95% CI, 9–17) months for CDC and RMC, respectively (Figure 1).

### 3.2. First-Line Treatment

Regarding first-line treatment, all patients received a platinum-based chemotherapy regimen. Nearly half of them received additional antiangiogenic treatment with bevacizumab, including 20 patients from the BEVABEL trial and 9 patients treated outside of a trial. The median duration of treatment was 4 months, and median TTP in the first line was 7.27 (95% CI, 7–10) months. ORR was 39% (95% CI, 26–52%), including 1 CR and 21 PR, and DCR was 67% (95% CI, 53–79) (Table 2).

### 3.3. Subsequent Treatments

Subsequent treatments included cytotoxic chemotherapy (*n* = 34), ICI (*n* = 20) and TKI (*n* = 12) (Table 3). A total of 36 patients (63%) received a second-line therapy (chemotherapy for 18 patients, ICI for 15 patients), and 22 patients (39%) received a third-line therapy. Chemotherapy regimens used were taxane (*n* = 14), platinum (*n* = 15), anthracycline (*n* = 3), and other chemotherapies (*n* = 2). For cytotoxic chemotherapy, the ORR and DCR values were 12% (*n* = 4/34) and 24% (*n*= 8/34), respectively.

In total, 20 patients received ICI, which included anti-programmed death-1 (PD-1)/programmed death-ligand 1 (PD-L1) monotherapy for all patients except for 2 patients who received an anti-PD-1 plus an anticytotoxic T-lymphocyte associated protein 4 (CTLA4). The ICI-associated ORR and DCR values were 10% (*n* = 2/20) and 30% (*n* = 6/20), respectively. In addition, 20 patients received TKI, including cabozantinib (*n* = 9), sunitinib (*n* = 2) or pazopanib (*n* = 1), with ORR and DCR values of 8% (*n* = 1/12) and 50% (*n* = 6/12), respectively. Other treatments included mTOR (*n* = 1), enhancer of zeste homolog 2 (EZH2; *n* = 1), proteasome (*n* = 1), and MEK (*n* = 1) inhibitors. No response was reported for these therapies. One patient whose tumor overexpressed MEK was treated with MEK inhibitors twice.

Notably, 9 patients with CDC were alive 2 years after diagnosis of metastatic disease. One of these patients received first-line treatment with CG + bevacizumab with the achievement of stable disease; the patient was free of treatment after 52 months of follow-up. Eight other patients received subsequent lines of therapy, including ICI (*n* = 6), chemotherapy (*n* = 7), and TKI (*n* = 4) (Table 4).

## 4. Discussion

To the best of our knowledge, this is the largest retrospective study reporting the efficacy of different treatments including ICI and antiangiogenic TKI as subsequent treatment lines in patients with metastatic CDC or RMC.

CDC and RMC are rare RCC subtypes with poor prognoses. Despite their renal location, these cancers differ from clear cell RCC, with a poorer prognosis and the presence of metastases common at diagnosis. In clinical trials, median overall survival in patients with stage 1–3 disease is approximately 50–70 months, while it is <12 months in those with stage 4 disease [6,7,8]. Until recently, most treatment approaches showed only limited activity. In the metastatic setting, platinum-based chemotherapy is considered standard first-line therapy; however, a phase 2 trial assessing cabozantinib, an antiangiogenic TKI, has recently reported promising efficacy results in the first-line setting. With an ORR of 35% and a median PFS of 6 months in 23 patients, cabozantinib could be considered a new option in this setting [5]. While these results are encouraging, nearly all patients will become resistant to first-line therapy, and currently, few data are available about subsequent treatments.

The characteristics of our population are consistent with data reported in the literature, with a male predominance and the presence of metastases at diagnosis. The median OS of 12.6 months was comparable with other studies; however, we did not find a difference in terms of mortality between CDC and RMC. In our cohort, patients received a median of two lines of treatment for metastatic disease, but one-third received only one line due to rapid progression. In the first-line setting, platinum-based chemotherapy was associated with an ORR of 39% and median TTP of 7.27 months, similar to the original trial by Oudard et al. [4].

There are currently no other studies (prospective or retrospective) reporting the efficacy of systemic therapies in subsequent lines of therapy in CDC and RMC; only individual case reports of patients treated with either targeted therapies or ICI are available. In this respect, our study is the first to assess treatment efficacy beyond the first line in patients with metastatic RMC or CDC. Our data suggest that subsequent treatments have only limited efficacy regardless of the therapeutic class used, with an ORR of 10–15%. However, better disease control was noted in patients treated with a TKI (mostly cabozantinib). Indeed, half of these patients achieved disease control with this treatment, mainly due to SD. Despite these slight differences, the duration of response for all treatments was extremely short, at around 2 months. While some studies have previously reported an efficacy signal with some TKIs [9,10,11], this signal was not observed in our study when TKI was administered as subsequent treatment; only one patient achieved an objective response, and five had SD with an extremely short duration of response. Regarding ICI, our results were similar to those with TKIs, contrasting with several case reports in which responses were observed [12,13,14]. One explanation for this might be the tumoral expression of PD-L1 in the patients treated with ICI in the published case reports. In our study, given its retrospective design and the date of the diagnosis (the oldest cases dating back to 2010), PD-L1 expression was not available and we were, therefore, unable to evaluate the response to ICI according to PD-L1 expression. Moreover, we already know that response to immunotherapy could be associated with other predictive biomarkers even in a PD-L1-negative population.

Nevertheless, we recorded 9 long survivors who lived more than 2 years, with three of them still alive after 40 months of follow-up. No further investigations such as next-generation sequencing or PD-L1 testing have been conducted in these patients, but these outcomes suggest intertumoral heterogeneity with varying evolution and prognosis between patients. Unfortunately, there are currently few data regarding the oncogenesis of these rare tumors, and the genomic profiles of CDC and RMC are still poorly understood.

A clear limitation of our study is its retrospective design, with its potential source of bias. Missing data, particularly in terms of molecular analysis, did not allow an analysis of treatment responses according to the molecular profiles of the tumors. Another limitation is the lack of a centralized pathologic review of these rare tumors that might be misdiagnosed. CDC and RMC differ especially in terms of epidemiology but also share many characteristics. A clinical analysis of 52 cases (39 CDC, 13 RMC) from a study group in Los Angeles highlighted an overlapping immunohistochemical profile [15]. Therefore, the histological definition of these two diseases is difficult to determine, due to, as seen before, overlapping forms from a molecular point of view but also due to a great heterogeneity of these tumors (CDC with loss of INI1 or fumarate hydratase expression) [16]. In addition, there are overlapping forms with other histological types. However, in our study, at least 20 tumors had a pathological central review as part of a clinical trial that confirmed the diagnosis of CDC or RMC (CARARE network). In spite of this, RMC and CDC were evaluated together in the clinical trial because they share anatomical and biological similarities and are currently treated similarly.

In light of our results, an understanding of the molecular biology of these tumors seems essential. The challenge of using genomic profiling to determine potential targets for treatment has already been studied. One study analyzed 17 cases of CDC using comprehensive genomic profiling. The most commonly observed genomic alterations included *NF2* (29% of tumors), *SETD2* (24%), *SMARCB1* (18%), and *CDKN2A* (17%). *PIK3CA, FBXW7, BAP1, DNMT3A, VHL*, and HRAS alterations were also identified (1/17 tumors had mutations in each of these genes). Putative therapeutic strategies based on these observed genomic alterations are mTOR pathway inhibition (*VHL, NF2, PIK3CA,* and *PIK3R2*), histone deacetylase inhibition (*FBXW7*), EZH2 inhibition (*BAP1* and *SMARCB1*), VEGF inhibition (*VHL*), and EGFR inhibitor resistance (*HRAS*) [17]. The discovery that *CDKN2A* was the most frequently altered gene in CDC samples is notable because alteration of *CDKN2A* has been associated with p16 deletion, which is responsible for overexpression of CDK4, TP53, and MYC. Indeed, p16 deletion may play a critical role in CDC biology through overexpression of oncogenic signaling pathways [18]. The increase in CDK4 could be a potential therapeutic target since selective inhibitors of CDK4/6 are now available and already widely used in the management of breast cancers.

Another study determined a unique transcriptomic profile in CDC tumors with alteration of immunogenic and metabolic pathways. These analyses showed a high enrichment of the immune signature in CDC. Examination of tumor slides for immune infiltration using CD3 and CD8 staining confirmed a high percentage of T-cell infiltrates in CDC. Importantly, the CD3 infiltration percentage was higher in metastatic tumors than in non-metastatic tumors; a similar trend was observed for CD8 cells. Moreover, a high percentage of CD8 cells in clear cell RCC is a poor prognostic biomarker (probably explained by the fact that lymphocytes are exhausted); it is also a predictor of response to immunotherapy [19]. In our trial, only a few responses were reported with immunotherapy agents, which may be explained by the enrichment of RMC in our study population. Indeed, RMC is known to be a poor responder to immunotherapy.

Other examples of case reports targeting specific abnormalities in the CDC have shown interesting results. For example, a human epidermal growth factor receptor 2 (HER2)-overexpressing mCDC treated with capecitabine and dual anti-HER2 blockade (trastuzumab, lapatinib) responded for more than 7 months with significant improvement in general condition [20]. With regard to RMC, most case studies evaluated cytotoxic chemotherapy agents [21,22,23].

The relatively common loss of *SMARCB1* in these tumors could provide a useful treatment target. Some studies have demonstrated the efficacy of EZH2 and AURKA inhibitors in patients with rhabdoid tumors, also defined by the loss of *SMARCB1* [24]. Furthermore, *SMARCB1* loss through the activation of the transcription factor MYC is involved in the proteasome pathway and autophagy, making these tumors sensitive to proteasome and autophagy inhibitors [25]. A phase 2 trial evaluating the efficacy of ixazomib combined with chemotherapy in patients with aggressive *SMARCB1*-deficient kidney malignancies is ongoing (NCT03587662).

Finally, the management of clear cell RCC is rapidly evolving, with several options in the first-line metastatic setting combining either dual ICIs or an ICI with a TKI. These combinations could also be considered in CDCs given the poor efficacy observed with mono-chemotherapy. This is particularly interesting since cases of response to ICI have been described. Monotherapy with an antiangiogenic TKI allows disease control in half of the patients but with a short median duration of response (2 months in our study). Therefore, a combination of both drugs might show promising results. Currently, two phase 2 trials are underway: one evaluating the efficacy of nivolumab and ipilimumab (Suniforecast) and the other evaluating pembrolizumab in combination with lenvatinib in advanced non-clear cell RCC (MSD MK 3475 B61).

## 5. Conclusions

To our knowledge, this is the largest study to date evaluating the efficacy of various treatments beyond the first line in patients with metastatic CDC or RMC. We found limited efficacy, regardless of the therapeutic class used, with a very short duration of response. Antiangiogenic TKI and ICI, although effective in clear cell RCC, did not show the same efficacy in these RCC subtypes. However, the presence of long responders who have been treated with ICI or TKIs suggests that there is a specific population that might benefit from these treatments. Additionally, new standards of treatment for metastatic clear cell RCC, combining antiangiogenic TKIs and ICI were not evaluated in this study and could be an interesting lead. All these data reinforce the idea that personalized medicine is mandatory for treating rare and aggressive cancers.

## Figures and Tables

**Figure 1 cancers-14-01678-f001:**
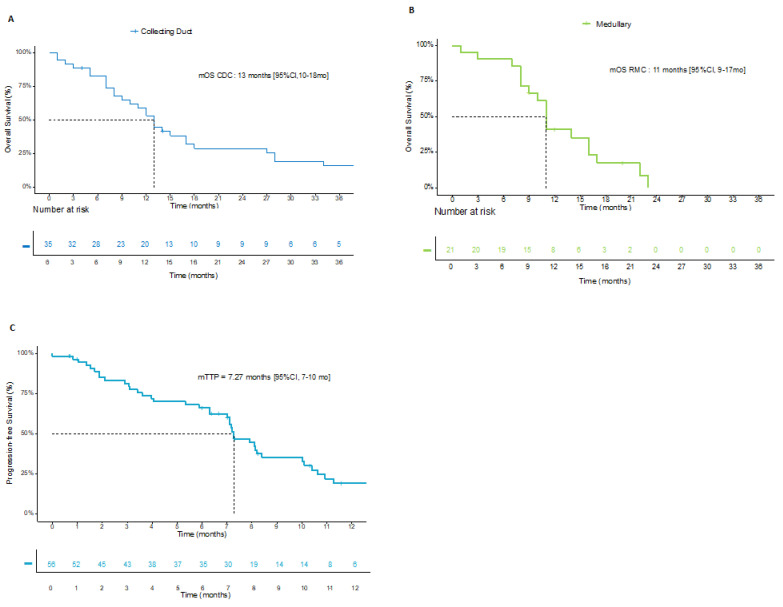
Survival curves (**A**) OS since metastatic stage in patients with CDC; (**B**) OS since metastatic stage in patients with RMC; (**C**) TTP in first line.

**Table 1 cancers-14-01678-t001:** Patient demographics and clinical characteristics.

Characteristic	CDC *n* = 35	RMC *n*= 22	Total Population *n* = 57
**Patient characteristics**			
**Median age (IQR), years**	61 (52–66)	33 (24–38)	53 (34–63)
**Male, *n* (%)**	23 (66)	17 (77)	40 (70)
**Disease characteristics**			
**Histology *n* (%)**			
CDC	35 (100)	0	
RMC	0	22 (100)	35 (61)
**Sickle cell traits, *n* (%)**	1 (3)	15 (68)	22 (39)
Missing data	14	3	16 (28)
***INI1* mutation, *n* (%)**			
Loss	3 (9)	15 (68)	18 (32)
Missing data	26 (74)	6 (27.2)	32 (56)
**Metastases at diagnosis, *n* (%)**	21 (60)	13 (59)	34 (60)
**Metastatic sites, *n* (%)**			
Lymph nodes	28 (80)	16 (73)	44 (77)
Bones	14 (40)	11 (50)	25 (44)
Liver	10 (29)	10 (45)	20 (35)
Lung	18 (51)	14 (64)	32 (56)
Other	15 (43)	9 (41)	24 (42)
**Pathologic stage at diagnosis *n* (%)**			
pT1–T2	5 (14)	1 (5)	6 (11)
pT3–T4	26 (74)	16 (73)	42 (74)
Missing data	4	5	9
**Lymph node invasion, *n* (%)**	19 (54)	15 (68)	34 (60)
Missing data	13	7	21
**Radical nephrectomy, *n* (%)**	27 (77)	13 (59)	40 (70)
**Median number of treatment lines (IQR)**	2 (1–5)	2.5 (1–4)	2 (1–5)
1 line (%)	46	23	37
2 lines (%)	23	27	24
≥3 lines (%)	31	50	39

**Table 2 cancers-14-01678-t002:** Clinical response with first-line therapies.

Type of Treatment	All Platinum-Based Chemotherapies *n* = 57	CG Regimen and Bevacizumab *n* = 29	CG Regimen *n* = 19	MVAC dd *n* = 9
**ORR, *n* (%)** Complete response Partial response Stable disease Progressive disease Missing data	22 (39) 1 (2) 21 (37) 16 (28) 8 (14) 8 (14)	12 (41) 0 12 (41) 11 (38) 2 (7) 2 (7)	5 (26) 0 5 (26) 5 (26) 4 (21) 4 (21)	5 (56) 1 (11) 4 (44) 0 2 (22) 2 (22)

**Table 3 cancers-14-01678-t003:** Efficacy of subsequent therapy lines.

Type of Treatment	First-Line Therapy	Subsequent-Line Therapy			
	Platinum-Based Regimen ± Bevacizumab *n* = 57	TKI ^a^ *n* = 12	ICI ^b^ *n* = 20	CT ^c^ *n* = 34	Other Treatment ^d^ *n* = 6
**ORR, *n* (%)** **DCR, *n* (%)** -Complete response -Partial response -Stable disease -Progressive disease -Missing data	22 (39) 38 (67) 1 (2) 21 (37) 16 (28) 8 (14) 8 (14)	1 (8) 6 (50) 0 1 (8) 5 (42) 3 (25) 3 (25)	2 (10) 6 (30) 0 2 (10) 4 (20) 14 (70) 0	4 (12) 8 (24) 0 4 (12) 4 (12) 20 (59) 6 (18)	2 (33) 2 (33) 0 2 (33) 0 4 (67) 0
**Median duration of treatment, months**	4	3	2	2	1

^a^ TKI: cabozantinib n = 8; sunitinib n = 1; pazopanib n = 1; ^b^ ICI: nivolumab n = 16; nivolumab + ipilimumab n = 2; ^c^ CT: taxane n = 14; platinum compound n = 15; anthracycline n = 3; other n = 2; ^d^ other treatment: MEK inhibitor n= 2; proteasome inhibitor n = 1; mTor inhibitor n = 1; EZH2 inhibitor n = 1; other n = 1.

**Table 4 cancers-14-01678-t004:** Characteristics of patients who achieved an objective response after subsequent-line therapy or were long-term survivors.

No.	First-Line Therapy	Best Response at First Line	Reason for First-Line Discontinuation	Second-Line Therapy	Best Response at Second Line	Reason for Second-Line Discontinuation	Third-Line Therapy	Best Response at Third Line	Reason for Third-Line Discontinuation	Overall Survival, Msonths
1	Cisplatin + gemcitabine + bevacizumab	SD	End of treatment	N/A	N/A	N/A	N/A	N/A	N/A	51
2 ^1^	MVAC dd	PD	Progression	Cisplatin + gemcitabine + bevacizumab	PR	End of treatment	Carboplatin + gemcitabine + bevacizumab	PR	Progression	30
3	Carboplatin + gemcitabine + bevacizumab	PR	End of treatment	Carboplatin + gemcitabine	PD	Progression	Nivolumab	PR	Still on treatment	43
4	MVAC dd	SD	End of treatment	Carboplatin + gemcitabine + bevacizumab	PR	Toxicity	N/A	N/A	N/A	69
5	Carboplatin + gemcitabine + bevacizumab	SD	Toxicity	Nivolumab	PD	Progression	Cabozantinib	PR	Toxicity	104
6 ^2^	Cisplatin + gemcitabine + bevacizumab	Dissociated	End of treatment	Trametinib	PR	Toxicity	Trametinib + crizotinib	PR	Progression	28
7 ^3^	Carboplatin + gemcitabine + bevacizumab	PR	Toxicity	Carboplatin	PD	Progression	Paclitaxel	SD	Progression	27
8	Carboplatin + gemcitabine + bevacizumab	SD	Toxicity	Nivolumab + ipilimumab	PD	Toxicity	Cabozantinib	SD	Progression	34
9	Carboplatin + gemcitabine + bevacizumab	PR	Toxicity	Nivolumab + ipilimumab	SD	Toxicity	N/A	N/A	N/A	48
10 ^4^	Cisplatin + gemcitabine + bevacizumab	PR	End of treatment	Nivolumab	PR	Progression	Cabozantinib	SD	Progression	22

^1^ Fourth line: paclitaxel + bevacizumab, fifth line: pazopanib; ^2^ fourth line: nivolumab; ^3^ fourth line: nivolumab, fifth line: gemcitabine; ^4^ fourth line: ixazomib.

## Data Availability

Patients were anonymized, and study documents were redacted to protect the privacy of trial participants.

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
