# Peer review of "Metastatic Renal Medullary and Collecting Duct Carcinoma in the Era of Antiangiogenic and Immune Checkpoint Inhibitors: A Multicentric Retrospective Study"

_cancers, 2022, doi:10.3390/cancers14071678_

Round 1

Reviewer 1 Report

I congratulate the authors on their efforts in presenting the retrospective review of two rare kidney cancers. Both CDC and RMC are rare, and this series will add to the existing literature about the challenges in the management of the metastatic disease.

Please review the grammar for the entire manuscript. Few examples:

LINE 62: Correct the sentence: CDC is mostly diagnosed…

Line 63 use “decade”

Can you clarify if all the 57 patients included in the study had metastatic disease (either denovo or recurrent after nephrectomy)

Do you have any data on stage II/III RMC or CDC treated with adjuvant therapy (chemotherapy, TKI, CPI) to see if it has any OS benefit?

Do you have any data on cytoreductive nephrectomy and survival outcomes?

The major limitation is the absence of genomic data in this patient series.

Reviewer 2 Report

This is a retrospective study to assess the efficacy of second-line or more therapy in patients with metastatic CDC/RMC. CDC and RMC are rare, so the results in the study is clinically informative for clinicians.

Major points

1) Figure 1 is not included in the manuscript. Only the legend of Figure 1 is included. In addition, survival curves of CDC and RMC should be presented separately.

2) I think that reader want to know detailed information of patients who achieved objective response after subsequent line or long-term survivors. So, it is better to present the baseline profiles and treatments of these patients as a table.

Minor

3) In the abstract, TTP should be described as time to progression (TTP). This term is firstly used.

Round 2

Reviewer 2 Report

I recommend acceptance of this paper.